# The Potential Systemic Role of Diet in Dental Caries Development and Arrest: A Narrative Review

**DOI:** 10.3390/nu16101463

**Published:** 2024-05-13

**Authors:** Ashley J. Malin, Zhilin Wang, Durdana Khan, Sarah L. McKune

**Affiliations:** 1College of Public Health and Health Professions, University of Florida, Gainesville, FL 32611, USA; z.wang6@ufl.edu (Z.W.); khan.durdana@ufl.edu (D.K.); smckune@ufl.edu (S.L.M.); 2College of Medicine, University of Florida, Gainesville, FL 32610, USA

**Keywords:** fat-soluble vitamins, dental caries, nutrition, children

## Abstract

Current conceptualizations of dental caries etiology center primarily on the local role of sugar, starch, or other fermentable carbohydrates on tooth enamel demineralization—a well-established and empirically supported mechanism. However, in addition to this mechanism, studies dating back to the early 1900s point to an important systemic role of diet and nutrition, particularly from pasture-raised animal-source foods (ASF), in dental caries etiology and arrest. Findings from animal and human studies suggest that adherence to a diet high in calcium, phosphorus, fat-soluble vitamins A and D, and antioxidant vitamin C, as well as low in phytates, may contribute to arrest and reversal of dental caries, particularly among children. Furthermore, findings from observational and experimental studies of humans across the life-course suggest that fat-soluble vitamins A, D, and K2 may interact to protect against dental caries progression, even within a diet that regularly contains sugar. While these historic studies have not been revisited in decades, we emphasize the need for them to be reinvestigated and contextualized in the 21st century. Specifically, methodologically rigorous studies are needed to reinvestigate whether historical knowledge of systemic impacts of nutrition on dental health can help to inform current conceptualizations of dental caries etiology, prevention, and arrest.

## 1. Introduction

Dental caries remains the most prevalent chronic disease in childhood and adulthood [1]. As of 2016, approximately 46% of youth aged 2–19 years had been diagnosed with dental caries [2]. Furthermore, 90% of US adults aged 20–64 years had either treated or untreated dental caries between 2011 and 2016 [3]. Dental caries is a multifactorial disease with behavioral, genetic, environmental, social, and policy-related influences [4,5]. For example, specific genes have been implicated in susceptibility to dental caries development, and dental care/hygiene practices, education, socioeconomic status, and country of residence as well as access to dental care have all been implicated in the development of dental caries [5,6,7]. Adequate nutrition is also essential for healthy tooth formation, and research shows that nutritional deficiencies can increase the risk of dental caries [8,9]. Furthermore, diet and nutritional adequacy can be influenced by education, socioeconomic status, country of residence and other sociocultural factors within the broader ecological system [10,11,12].

Current conceptualizations of dental caries etiology center primarily on the local or topical role of diet in the oral cavity; particularly, the effects of sugar, starch, or other fermentable carbohydrates on tooth enamel demineralization [13]. However, in addition to this mechanism, there are peer-reviewed studies dating back to the early 1900s that highlight an important systemic role of diet and nutrition, particularly from pasture-raised animal-source foods (ASF), in dental caries etiology and arrest (see Table 1 for a summary of these studies). Despite the apparent prominence of this understanding at the time and the public health relevance of these studies today, they have not been revisited by the scientific community for decades. The following paper provides a comprehensive review of these historic studies and highlights the importance of reinvestigating whether this historical knowledge can help to inform current conceptualizations of dental caries etiology, prevention, and arrest. Furthermore, this review discusses potential mechanisms by which nutritional factors from ASF might systemically impact dental caries etiology and arrest.

## 2. Vitamins A and D

Vitamin A and vitamin D are fat-soluble vitamins that have been implicated in the development of dental caries. Vitamin D plays an integral role in plasma calcium and phosphorus regulation and is essential for tooth and bone mineralization; while vitamin A is associated with dental and skeletal growth [32,33]. Research shows that deficiencies in vitamin A and vitamin D can disrupt salivary flow or interfere with protective qualities of saliva, thereby increasing the risk of caries. Furthermore, severe vitamin D deficiencies have been shown to contribute to tooth hypo-mineralization and enamel hypoplasia [34,35]. Lower levels of vitamins A and/or D have been associated with higher prevalence of dental caries among adults and/or children in some epidemiological studies [36,37,38,39,40,41,42] but not others [43]. It has been suggested that vitamin D may protect against dental caries by modulating immunity in a manner that promotes eradication of cariogenic bacteria [36]. However, findings from studies conducted in the early 1900s suggest that vitamins A and D may promote resistance to dental caries via their impacts on tooth structure and enamel mineralization.

In the early 1900s, Drs. Edward and May Mellanby conducted experimental studies on dogs and rabbits that examined effects of vitamin A, vitamin D, calcium, and phosphorus intake on tooth formation and resistance to decay. They found that puppies fed animal fats rich in vitamin A and D, such as cod liver oil, butter, and suet had healthy tooth development; whereas those fed linseed oil, devoid of these vitamins, had poor tooth development [14]. Furthermore, they showed that puppies fed a diet deficient in vitamin A but otherwise nutritionally complete, had low tooth calcium content, delayed loss of deciduous teeth, delayed eruption of permanent teeth, abnormal positioning and overlapping of teeth, and defective enamel [14]. Consistently, Agnew et al. (1933) found that rats fed a diet low in phosphorus and vitamin D for at least two months developed dental caries. However, in that study the diet was also low in other minerals and quality proteins, and therefore, the role of phosphorus and vitamin D could not be parsed [28]. Subsequent research by Grieves (1922) demonstrated that rats fed a diet deficient in calcium, protein, and vitamins A and D developed dental caries, while those fed a diet sufficient in these nutrients did not [16]. He concluded that the relative proportion of these nutrients was most important for dental health, rather than any one of them alone [15].

## 3. Anti-Nutrients and Vitamins A and D

Consuming a diet rich in plant-based foods can help to prevent and reduce the risk of chronic illnesses [44,45]. However, plant-based foods can also exert harmful health effects because they contain “anti-nutrients”. These anti-nutrients are part of the plant’s defense system but can also interfere with nutrient absorption and bioavailability [46,47]. Common anti-nutrients include phytates, tannins, oxalates, goitrogens, phytoestrogens, lectins, and saponins [46,48,49]. Consumption of anti-nutrients may interfere with nutritional status, thereby impacting dental health; however, their potential role in dental caries development has received little attention in the contemporary literature.

### 3.1. Animal Studies

In an experimental study from 1926, M. Mellanby provided rabbits with diets varying in the calcium to phosphorus ratio, as well as vitamin D, vitamin A, and phytate (i.e., cereals) quantities. Rabbits fed a diet of crushed oats, bran, and lemon juice that was supplemented with calcium carbonate and cod liver oil or egg yolks had normal tooth development and healthy tooth calcification [19]. Conversely, rabbits fed the same diet but without the cod liver oil or egg yolks had poor tooth calcification. Interestingly, vegetable consumption did not appear to contribute to the reported “calcifying effects” observed from ASF consumption; however, of all the vegetables examined, green vegetables appeared to be most facilitative of calcification [19]. Furthermore, in a similar study conducted among puppies, Mellanby (1928, 1929) reportedly found that when cereals, particularly oatmeal, constituted a significant proportion of the diet compared to white rice or flour, teeth exhibited noticeable “anti-calcifying effects” unless vitamin D was also abundant in the diet [25,50]. Interestingly, unlike whole grains, when white rice or flour are consumed, the germ containing the anti-nutrient component is removed. M. Mellanby (1926, 1928) concluded that whole grains contain an “anti-calcifying” component that can contribute to the development or worsening of rickets as well as delays in tooth calcification; however, these effects can be ameliorated if vitamin D is abundant in the diet [19,50]. Consistent with the Mellanby (1918, 1926, 1928) animal studies, relatively recent research shows that phytates (i.e., phytic acid) present in whole grains, seeds, legumes, and nuts, can decrease absorption of iron, zinc, magnesium, and calcium, which can contribute to bone demineralization [51,52]. Additionally, lectins, saponins, and tannins, which are all anti-nutrients, have been shown to interfere with nutrient absorption [49].

### 3.2. Child Studies

M. Mellanby et al. also conducted studies among school-aged children on the potential role of vitamins A and D in dental caries etiology and arrest. In one study, the children (aged 7–7.5 years old) resided in institutions and had poorly developed teeth prone to demineralization and oral health issues [17]. Like her animal studies, in Mellanby et al. (1924) [17] children were provided with diets differing in calcium, vitamin D, and phytic acid content. They were assigned to one of three groups, and their dental health was assessed on average 7.5–8 months later. Group one (*n* = 9) had a diet abundant in fat-soluble vitamins and calcium and lower in phytic acid (i.e., extra milk, eggs, and cod liver oil, as well as no oatmeal); Group 2 (*n* = 10) had a diet with less calcium and fat-soluble vitamins (i.e., “very little egg”, less milk, no cod liver oil) than Group 1 and it included oatmeal; Group 3 (*n* = 13) consisted of the ordinary hospital diet that had foods containing quantities of calcium, fat-soluble vitamins, and phytic acid intermediate between Groups 1 and 2. The findings showed that the average number of teeth per child that showed initiation or spread of caries was 1.4, 5.1, and 2.9 in each of the three groups respectively, while the average number of teeth per child in which the caries showed hardening (i.e., improvement) was 1.5, 0.7, and 1.0 for each group respectively. Conversely, the average number of teeth per child in which dental caries showed softening (i.e., worsening) was 0, 0.4, and 0.1 for each group respectively. Therefore, consistent with M. Mellanby’s animal study findings, these findings suggest that the diet most abundant in calcium and fat-soluble vitamins, and lowest in phytic acid, had the most restorative impact on dental caries. Conversely, the diet with the least calcium and fat-soluble vitamins, as well as highest in phytic acid, appeared to be the most dental caries-promoting [17]. Of note, the sample size in this study is small, and children were institutionalized, which may limit generalizability of the findings.

In later research, Mellanby & Pattinson (1926) [20] included larger sample sizes and kept the total energy, calcium–phosphorus ratio, and acid–base ratios in the children’s diets constant. They also attempted to keep the protein, carbohydrate, and fat intakes constant, varying only the quantities of fat-soluble vitamins and cereals [20]. Group 1 (*n* = 23) was provided with a diet abundant in fat-soluble vitamins, Group 2 (*n* = 24) had significantly less fat-soluble vitamins than Group 1, as well as oatmeal substituted for some bread, and Group 3 (*n* = 24) had quantities of fat-soluble vitamins intermediate between Groups 1 and 2 with no oatmeal. The average number of teeth per child observed to have initiation or spread of caries after six months on the diet was 1.8, 5.8, and 3 for each of the three groups respectively. Conversely, the average number of teeth per child in which dental caries showed hardening (i.e., improvement) was reported to be 2, 0, and 1.2 for Groups 1, 2, and 3 after six months, respectively [20].

Mellanby & Pattinson (1928) conducted a subsequent study that examined effects of vitamin D supplementation on dental caries progression among institutionalized children under the age of six [21]. There were four groups of children included (per group, *n* =18–21). The authors utilized data from children under the age of six from Mellanby (1924, 1926) such that groups A1/A2, B1/B2, and C1/C2 had the same diets as Groups 1, 2, and 3 from these studies respectively. Group A3 was also added, which included the standard institutional diet plus irradiated ergosterol in the form of radiostol as a source of vitamin D. It is notable that diet A3 also contained 1 oz. of sugar and 1 oz. of syrup or jam per day. Children in this group had an average of 8.8 cavities at the start of the study. The findings showed that children in Group A3 had slightly fewer teeth showing initiation or spread of caries (1 tooth compared to 1.4 teeth), and slightly more teeth showing hardening or arrest of caries (3.9 teeth compared to 3.7 teeth, respectively) than Group A1/A2 (i.e., the group with the diet abundant in fat-soluble vitamins but not supplemented with additional vitamin D) after 28 weeks. Furthermore, both groups A1/A2 and A3 had fewer teeth showing initiation or spread of caries and more teeth showing hardening or arrest of caries than groups B1/B2 or C1/C2 (i.e., those with diets lower in fat-soluble vitamins) [21]. These findings again highlight the potential importance of fat-soluble vitamins, particularly vitamin D, for dental health among children.

An interesting component of Mellanby & Pattinson (1928) is that they published photomicrograph images of what they considered to be the caries-arresting process. They provided an example of a participant for whom secondary dentin appeared to have formed, and the soft area of the caries-infected tooth hardened, after following the nutritional protocol [21]. Furthermore, Mellanby (1947) subsequently stated “if a thin section of such a tooth is examined under the microscope it will in most cases be seen that a large barrier of newly formed dentine has been deposited between the primary dentine and the pulp in the region of the original carious attack. This so-called secondary dentine is usually well calcified” ([53], p. 17). These findings suggest that regrowth of dentin may be evident following adherence to a diet rich in fat-soluble vitamins.

Based on the collective findings from these child studies, Mellanby & Pattinson (1932) concluded that a diet high in vitamin D and calcium but low in cereals has important protective and restorative effects on dental caries [25]. They postulated that grain consumption counteracts the positive effects of vitamin D (what they referred to as a “calcifying vitamin”) in preventing or slowing the progression of caries among children [21,25]. Interestingly, it was these findings in part that led the United Kingdom (UK) to fortify wheat flour (except whole meal and some self-raising varieties) with calcium in 1943 [47,54,55]. This is the only mandatory program of calcium fortification worldwide [54]. Furthermore, in 1939 May Mellanby and her husband, scientist Edward Mellanby, were nominated for a Nobel Prize in Physiology or Medicine for their work on dietary deficiencies and human disease. For her research on nutrition and dental caries, Göran Liljestrand, secretary of the Nobel Committee, referred to her research as “having been of great importance for humanity” and “having revolutionized dental science” [56]. Though ultimately not awarded the Nobel Prize, historical researchers have speculated that gender bias may have played a role in this outcome, as May Mellanby was the only woman shortlisted for the prize, and the scientific community contributed to the final decision that her work be considered in conjunction with a joint nomination of her husband [56].

Similar studies conducted by other researchers during the 1920s and 1930s report findings consistent with M. Mellanby. For example, Anderson et al. (1934) conducted a study of 162 children aged 2–16 years residing in two orphanages in Toronto, Canada [29]. Children were divided into two groups and their diets kept nearly identical except that one group was provided with eight drops of 250 D viosterol daily. In both groups, their diets contained meat, eggs, milk, fruits, and vegetables, as well as cereal. The findings showed that the number of new cavities in deciduous and permanent teeth per child was less than half in the group provided with vitamin D compared to those that did not receive vitamin D supplementation. Furthermore, the number of “markedly progressive caries” was reportedly lower in the group supplemented with vitamin D, while the number of “non-progressive caries” was higher in the vitamin D supplemented group over a 1-year period [29].

In the 1940s, Mellanby and Coumoulos (1944) published a retrospective epidemiological study that examined dental health among two separate cohorts of over 1000 5-year-old children in London, England, in 1929 and in 1943 [30]. During this timeframe, several public health nutrition changes were implemented for pregnant and nursing women as well as children, in addition to calcium carbonate being added to bread. These included lowered cost of milk, cod liver oil, and orange juice, as well as increased availability of eggs for pregnant and lactating women, infants, and young children. Furthermore, as part of the government’s wartime food policy, most schools began providing higher-quality dinners to anyone interested. Lastly, vitamin D2 and vitamin A were added to all margarine [30]. Findings showed that among 1571 children in 1929 and 1139 children in 1943, the incidence of having “much” dental caries in deciduous teeth decreased from 62.8% to 29.3%, while the incidence of being “caries free” increased from 4.7% to 22.4% between the two timepoints [30]. Mellanby & Mellanby (1948) followed additional cohorts of 5-year-old children over subsequent years and reported even greater improvements in their dental health over time [31].

## 4. Antioxidant Vitamins C and E

Emerging evidence suggests that higher levels of antioxidants vitamin C and vitamin E in saliva may be associated with decreased risk of caries. Specifically, recent cross-sectional studies of children in India and Saudi Arabia have found significantly lower salivary vitamin C and/or E among children with active dental caries [57,58]. Research from the early 1900s also points to a potential protective effect of vitamin C on dental caries progression.

Howe (1920) [15] demonstrated that feeding Guinea pigs a diet of oatmeal and skim milk produced dental caries while the addition of orange juice arrested them. Furthermore, after the removal of orange juice from the diet, dental caries reemerged, while the addition of sugar to the diet did not impact caries incidence if the diet was otherwise nutritionally adequate [15]. Thus, in contrast to the Mellanby and Pattison studies, Howe’s findings highlight the importance of vitamin C for buffering potential cariogenic effects of sugar and poor nutrition. In subsequent research, Howe (1924) conducted studies with monkeys in which he varied the amounts of calcium and vitamin C and concluded that it is the interplay between nutrients in the diet that is most important for dental health. He emphasized that calcium, phosphorus, vitamin C, and protein are especially important in dental disorders [15,18].

Subsequent human studies by other scientists further emphasize the importance of vitamin C for dental health. Hanke (1930) reported that among 61 participants aged 0–50 years who had caries, all were deficient in vitamin C, and among 39 of these participants, this was their only “demonstrable” nutritional deficiency [24]. Furthermore, in a subsequent experimental study, Hanke (1933) provided 323 institutionalized children and adolescents aged 10–17 with a pint of orange juice and juice from one lemon per day for one year in addition to the standard institutional diet. He observed that approximately 50% of participants experienced arrest of their existing dental caries [26].

## 5. Observational and Experimental Studies on Vitamins A, D, and K2 across the Life Course

Around the same time that M. Mellanby and others were conducting experimental studies on nutrition and dental caries, Dr. Weston A. Price (a Canadian dentist and founder of the National Dental Association) was conducting observational research on this topic. Dr. Price travelled the world examining nutritional practices of different cultures. His primary interest was in facial structure development and factors contributing to tooth decay resistance among people who have low or no prevalence of dental caries [59]. He journeyed across the world, visiting indigenous people from every continent during the 1920s and 1930s. His decades-long journey culminated in his book “Nutrition and Physical Degeneration”, published in 1939. Throughout his travels, he observed potential evidence that dental caries as well as orthodontic issues may be caused by nutritional deficiencies [59]. Furthermore, his research led him to discover a “fat-soluble activator” that we now know to be a form of vitamin K2. His observations suggest that consumption of foods rich in vitamin K2 (e.g., ASF such as grass-fed raw dairy, as well as eggs and organ meats of animals) may have dental health benefits. Specifically, he observed that switching from traditional diets rich in calcium and phosphorus as well as vitamins A, D, and K2 to modern diets resulted in physical degradation, including an increase in dental caries. In contrast, he found that people who followed their traditional diets were virtually free of dental caries [59].

Some of Dr. Price’s most notable observations were those among Alaskan indigenous populations. In one settlement, he studied 12 individuals, 10 of whom consumed exclusively or almost exclusively traditional local food. Of their 288 teeth examined in total, only one tooth (i.e., 0.3% of the teeth examined) was observed to have been affected by decay. Conversely, among the two individuals who regularly consumed store-bought foods, 27% of their teeth examined had caries. In another settlement, among 88 people observed, 27 nearly exclusively consumed traditional food and had only one tooth amongst a total of 796 teeth examined affected by decay (i.e., 0.1% of teeth examined). Conversely, among the 40 people in this settlement who subsisted entirely on modern food brought in by government supply ships, 21.1% of their 1094 teeth examined had cavities. Among the 21 people who ate both traditional and store-bought foods, 38 of their 600 teeth examined or 6.3% had decay. Interestingly, the foods comprising the traditional diet of the Alaskan indigenous populations consisted largely of marine life; especially fish, seals, and whales, which are high in protein, fat, OMEGA-3 fatty acids, and vitamin D. Furthermore, indigenous people in some areas consumed a combination of seafood, as well as berries, roots, and wild animals, which are also nutritionally dense [59].

There are numerous other examples in Price (1939) describing superior dental health and virtual absence of dental caries among indigenous peoples who followed their traditional nutrient-dense diets. Notably, Price (1939) observed that those following their traditional diets often consumed their ASF raw and unprocessed, which may help to preserve nutrient density. Additionally, they tended to utilize fermentation for food preservation of plant-sourced foods. This dietary preservation method may also effectively retain nutrients in the food and favorably impact overall health, including dental health [59].

Dr. Price also conducted experimental studies showing that fat-soluble vitamins A, D, and K2 may interact to protect and improve dental health. He described his “reinforced” nutritional program to treat dental caries, which included liberal amounts of foods high in vitamins A, D, and K2. It consisted of adding a mixture of half of a teaspoon of high-vitamin cod liver oil (rich in vitamins A and D) and high-vitamin butter oil (rich in vitamin K2) three times per day to meals low in sugar and starches. He noted that this mixture was more effective than either cod liver oil or butter oil alone [59]. Indeed, many of the indigenous cultures that Dr. Price observed to be virtually caries-free regularly consumed raw dairy from grass-grazing animals, seafoods, and organ meats abundant in these fat-soluble vitamins. Dr. Price postulated that his reinforced nutritional protocol helps to remineralize teeth by improving the quality of saliva [59]. He specifically noted, “available data indicate that the blood and saliva normally carry defensive factors which when present control the growth of the acid producing organisms and the local reactions at tooth surfaces” ([59] p. 225). Indeed, some studies from the 1920s and 1930s show associations of calcium and phosphorus levels in blood and saliva with dental caries, although findings are inconsistent [59,60,61,62,63,64,65].

Price (1939) [59] reports a study of 17 patients between the ages of 12 and 22 years who were provided with his “reinforced” nutrition protocol during the winter and spring months. These patients were followed up every 6–12 months over three years. He reported that results of X-rays and clinical examinations showed that prior to beginning the protocol, approximately half of the teeth examined had open cavities, while after three years on the protocol only two new cavities developed in the group. He replicated these findings in a subsequent study with a sample size of 50 [59]. Furthermore, he conducted an experimental study during the industrial depression among school-aged children of mine workers whose home meals were nutritionally deficient. He provided them with a nutritionally “reinforced” meal for lunch six days per week while the home meals and home dental care routines remained unchanged. Regarding the nutritionally reinforced meal, he noted “about four ounces of tomato juice or orange juice and a teaspoonful of a mixture of equal parts of a very high vitamin natural cod liver oil and an especially high vitamin butter was given at the beginning of the meal. They then received a bowl containing approximately a pint of a very rich vegetable and meat stew, made largely from bone marrow and fine cuts of tender meat: the meat was usually broiled separately to retain its juice and then chopped very fine and added to the bone marrow meat soup which always contained finely chopped vegetables and plenty of very yellow carrots; for the next course they had cooked fruit, with very little sweetening, and rolls made from freshly ground whole wheat, which were spread with the high-vitamin butter. The wheat for the rolls was ground fresh every day in a motor driven coffee mill. Each child was also given two glasses of fresh whole milk. The menu was varied from day to day by substituting for the meat stew, fish chowder or organs of animals” ([59], p. 217). Conversely, the home diet consisted of “highly sweetened strong coffee and white bread, vegetable fat, pancakes made of white flour and eaten with syrup, and doughnuts fried in vegetable fat” ([59], p. 217). He reported an improvement in the quality of the children’s saliva associated with the cessation of tooth decay in as little as six weeks, with continued improvements over five months [59]. While Weston Price’s research findings are limited by a lack of scientific peer review, and in some cases vaguely described observations, they are consistent with findings from peer-reviewed studies conducted by other scholars during that era.

For example, Boyd & Drain (1928) reported that 28 children who had been under medical control and nutritional management for diabetes for at least six months exhibited arrest of their dental caries [22]. Interestingly, 82% of these children reportedly had progressive dental caries prior to starting the nutritional protocol. The nutritionally managed diet included eggs, butter, meat, cod liver oil, fruits, bulky vegetables, and 1 quart of milk and cream per day, along with insulin [15]. In a small subsequent study of four children without diabetes who were also provided with the same nutritionally managed diet, they found that reverting from the nutritionally managed diet to the original diet contributed to reemergence of caries in two children. Thus, they attributed the arrest of dental caries among diabetic children to improvements in dietary nutrition rather than the provision of insulin [15,23], although the sample size is quite small. Boyd et al. (1929) also reported arrested caries among small groups of children after the provision of diets that included “a quart of milk, one egg, a teaspoon of cod liver oil, one ounce of butter, one orange, two or more servings of succulent vegetables or fruits, and such other foods as the child desired”, even if the diet included candy after meals; noting that caries arrested after 2.5 months in this case [23]. Moreover, they reportedly observed arrested caries after more than three years among children with celiac disease who were provided with a variation of this dietary protocol that included more sugar and simple carbohydrates, even though oral hygiene practices remained unchanged [15,23]. Consistently, Howe, White, and Rabine (1933) reported that the following diet was recommended to children aged 2–11 who were patients at the Forsyth Dental Infirmary: “1 quart (946 cc.) of milk; one raw vegetable (or canned tomato), with special emphasis on cabbage and tomato; at least two cooked vegetables, one of which may be potato; two servings of fruit, with special emphasis on oranges; one egg; meat or fish five times a week, and butter on vegetables and bread. An attempt was made to keep cereals and breads, because of their acid ash, as low as was consistent with the requirements for energy. Candy was allowed only at the end of the meal. Cod liver oil was not given as a routine but was prescribed by the pediatrician in occasional cases” ([27], pp. 1048–1049). Their findings showed that among 132 patients who reported compliance with the protocol, dental caries reduced by approximately 79% in just over one and a half years. These findings, along with Weston Price’s, not only highlight the potential for a diet rich in vitamins A, D, and K2 (and in some cases, low in phytates) to contribute to the arrest of dental caries, but also for it to do so in the context of a diet that regularly contains sugar.

While the mechanism(s) by which vitamin K2 may protect the teeth from adverse effects of sugar consumption have not been investigated, Southward (2015) hypothesized that it may do so by increasing the antioxidant capacity of the hypothalamus [66]. He stated that the hypothalamus regulates centrifugal fluids in the teeth by signaling the parotid gland to release saliva which nourishes, protects, and helps to clean teeth. He hypothesized that high sugar intake may increase reactive oxygen species and oxidative stress in the hypothalamus, which can cause this signaling mechanism to stop or even for dentinal fluid to flow in reverse [66]. As the flow of centrifugal fluids in the mouth slow down or stop, oral bacteria can more easily attach to the surface of the tooth and contribute to dental erosion [66]. Indeed, research shows that antioxidants may help to control free radicals in the hypothalamus caused by blood sugar spikes, and vitamin K2 has been shown to have neurological antioxidant effects [67,68]. While Dr. Ken Southward’s hypothesis has not yet been empirically tested, it has been recognized by other scholars [69,70,71]. Vitamin K2 consumption may also help to improve tooth mineralization via its impacts on enzymatic expression. Vitamin K2 acts as a co-factor of enzyme ‘vitamin K-dependent carboxylase’ which can alter the structure of proteins via gamma-carboxylation. Vitamin K2 has been shown to improve gamma-carboxylation of osteocalcin, a protein found in bones and teeth. Moreover, osteocalcin requires fat-soluble vitamins A and D in its production [72].

To our knowledge, there are no contemporary studies examining interactions between vitamins D and K2 in relation to dental health; however, findings from two relatively recent experimental studies show that these vitamins can interact synergistically to improve bone density. Iwamoto et al. (2000) found that among 92 women aged 55–81 years with osteoporosis, those provided with vitamins K2 and D3 showed significant increases in bone mineral density after one year that were significantly greater than those administered vitamins K2, D3, or calcium alone [73]. In a subsequent study, Ushiroyama et al. (2002) also found that the provision of vitamins K2 and D3 together for 24 months markedly improved bone mineral density in a study of 172 postmenopausal women with osteopenia and osteoporosis; whereas improvements from vitamin K2 alone were significantly less [74]. However, it is unknown whether food-based rather than supplement-based consumption of vitamins D and K2 might differentially impact bone density.

## 6. Limitations of Historic Studies

While the historic studies reviewed herein provide compelling and consistent evidence of a potential systemic role of nutrition in dental caries etiology and arrest, they also have limitations. First, several of the initial M. Mellanby animal and human studies have very small sample sizes [14,17]; while her subsequent human studies employed larger sample sizes, these still included institutionalized children [20,21] for which findings might not be generalizable to non-institutionalized children. Some of these publications also lack key methodological details pertaining to experimenter masking, participant selection, and/or control/assessment of child health status which could impact dental caries progression [20,21]. However, subsequent studies with larger sample sizes conducted by other researchers reportedly obtained comprehensive health histories [24,27]. The observational studies conducted by Mellanby & Pattinson are also limited in that they did not longitudinally follow the same groups of children over time; rather, they compared separate cohorts of 5-year-old children at different timepoints. Therefore, they could not assess dental caries prevalence within participants over time. Studies conducted by Boyd & Drain also included small sample sizes and were quasi-experimental, examining dental caries progression among children under nutritional management while apparently lacking a comparison group [22,23]. However, subsequent studies by other researchers that included relatively large sample sizes did include comparison groups [24,27]. Lastly, a limitation when considering findings from all the historic studies is that methods for assessing and diagnosing dental caries have greatly advanced since these studies were published [75,76,77,78]. Therefore, studies employing modern methods for assessing dental caries are needed to examine research questions tested in the historic studies described herein.

## 7. Discussion and Future Directions

Scientific conceptualizations around dental caries etiology have evolved considerably over time [79]. In the 1700s, dental caries was considered to develop from inflammation of the lining between the dentin and tooth pulp, and in the late 1700s to 1800s, the role of bacterial fermentation was discovered [79]. Still, the innate ability of teeth to withstand the process of demineralization under certain conditions was also acknowledged. In 1848, dentist-microscopist Tomes stated that “the dentinal tubule complex contained a life force by which the dentin was able to build a barrier against the process of disintegration and that dentine is possessed of vitality… and that vitality must have been lost before caries began…” [79]. By the 1930s, a scientific consensus appeared to have been reached that diet and nutrition served an integral systemic role in dental caries etiology and arrest. This research culminated in a Nobel Prize nomination for one of the most notable scholars of that era, Dr. May Mellanby; yet, from the 1950s onward, research on nutrition as an intervention for dental caries dissipated. While relatively recent books by dentists [80] and parents [81] have based nutritional protocols for healing dental caries on some of the research reviewed in this paper, this fascinating area of research has otherwise virtually been lost in time. Current research on dental caries etiology and remineralization centers primarily on the topical role of sugar in bacterial fermentation in the oral cavity, a well-established and empirically supported mechanism [82,83] with importance given the prevalence of sugar in the modern diet. However, the consistency in the findings of the nutrition studies included in this review is striking, and their public health implications potentially (re)revolutionary. There is a significant need for this research to be contextualized in the 21st century.

Dental health and diet are increasingly linked to overall health and influenced by various components within the overarching ecological system, including public policy, education, country of residence, and socioeconomic status [4,6,7,11,12]. ASF consumption is contentious, as it clearly has environmental implications [84]; however, while many developed countries consume ASF on a regular basis, thus meeting their nutritional needs, much of the population of low- and middle-income countries lack access to and regular consumption of ASF [85,86]. Moreover, much of the ASF consumed is produced in such a way (i.e., often via factory farming) that it has significant negative health and planetary implications. These include increased risk of chronic and infectious disease, as well as deforestation and increased carbon emissions [87,88]. Nevertheless, given the public health implications of findings from studies reviewed herein, the potential dental health benefits of a diet rich in ASF must be probed with consideration for balancing environmental and food production factors.

A limitation of this review is that none of the historic studies included conducted randomized controlled trials. Methodologically rigorous experimental and prospective cohort studies are needed to examine whether a diet abundant in calcium, phosphorus, and fat-soluble vitamins, and low in phytates, has beneficial effects on dental caries incidence and progression. Given that the focus of the historic studies was on children, we similarly suggest that contemporary future studies focus on children but with consideration for the entire life-course. Furthermore, future studies should control or adjust for individual (e.g., health status, dental hygiene), social (e.g., education, socioeconomic status, access to dental care) and public health (e.g., access to ASF and/or fortified foods, ASF food production) factors that may influence diet and dental health. Lastly, these studies should undoubtedly employ the most rigorous contemporary methods for dental caries assessment and diagnosis.

## 8. Conclusions

The systemic impacts of a diet high in beneficial minerals and fat-soluble vitamins as well as low in phytates should be reinvestigated as an additional mechanism in the etiology and arrest of dental caries. Furthermore, given the limitations noted in the studies included in this review, methodologically rigorous studies that employ modern approaches for dental caries assessment are needed.

## Figures and Tables

**Table 1 nutrients-16-01463-t001:** Summary of Historic Studies on Systemic Impacts of Nutrition on Dental Caries.

	Title	Year	Authors	Participants/Subjects	Design	Foods/Nutrients Examined	Main Findings
1	An Experimental Study of the Influence of Diet on Teeth Formation	1918	M. Mellanby [14]	8-week-old puppies N = 3	Animal	Cod liver oil, butter, linseed oil; vitamin A	A diet abundant in fat-soluble vitamin A is associated with healthy tooth development; a diet deficient in vitamin A is associated with dental defects.
2	Dental Caries ^a^	1920	P.R. Howe [15]	Guinea pigs	Animal	Oatmeal, skim milk, orange juice	A diet of oatmeal and skim milk produced dental caries while the addition of orange juice arrested them. The addition of sugar did not impact caries incidence if the diet was nutritionally adequate.
3	A Preliminary Study of Gross Maxillary and Dental Defects in Three Hundred Rats on Defective and Deficient Diet ^a^	1922	C.J. Grieves [16]	Rats N = 300	Animal	Calcium, protein, and vitamins A and D	Rats fed a diet deficient in calcium, protein, and vitamins A and D developed dental caries; those fed a diet sufficient in these nutrients did not.
4	The Effect of Diet on the Development and Extension of Caries in the Teeth of Children	1924	M. Mellanby, C.L. Pattison, J.W. Proud [17]	7–7.5 yr. old children residing in institutions N = 32 (*n* = 9–13 per each of the three groups)	Human/ Experimental	Calcium, vitamin D, and phytic acid; milk, cod liver oil, eggs, oatmeal	Adhering to a diet abundant in fat soluble vitamins and calcium and lower in phytic acid for 7.5–8 months was associated with lower incidence of caries initiating or spreading and more caries hardening compared with a diet lower in fat-soluble vitamins/calcium and higher in phytic acid.
5	Studies of Dietary Disorders Following Experimental Feeding with Monkeys	1924	P.R. Howe [18]	Monkeys	Animal	Calcium and vitamin C	Limiting dietary calcium or vitamin C contributed to dental caries.
6	A Preliminary Study of Factors Influencing Calcification Processes in the Rabbit.	1926	M. Mellanby & E.M. Killick [19]	8-week-old rabbits	Animal	Cod liver oil, egg yolks, oats, bran, lemon juice, savoy cabbage, other vegetables; calcium to phosphorus ratio; vitamins A and D; phytates	Rabbits supplemented with cod liver oil or egg yolks had normal tooth development and tooth calcification; those not supplemented with cod liver oil or egg yolks had poor tooth calcification.
7	Some Factors of Diet Influencing the Spread of Caries in Children.	1926	M. Mellanby, C.L. Pattison [20]	8.7–9-year-old children N = 71 (*n* = 23–24 per each of the 3 groups)	Human/ Experimental	total energy, calcium-phosphorus ratio, acid-base ratio, protein, carbohydrates, fat; fat-soluble vitamins and cereals	After 6-months on the diet, the diet most abundant in fat-soluble vitamins and lower in phytic acid was associated with lower incidence of caries initiating or spreading and with more caries hardening than a diet lower in fat-soluble vitamins and higher in phytic acid.
8	The Action of Vitamin D In Preventing the Spread and Promoting the Arrest of Caries In Children	1928	M. Mellanby, C.L. Pattinson [21]	Institutionalized children under the age of 6; N = 78; *n* = 19–21 per each of the 4 groups	Human/ Experimental	Calcium, vitamin D, phytic acid; irradiated ergosterol as vitamin D supplementation	The addition of vitamin D supplementation to the diet contributed to greater hardening of caries and lower initiation/spread of caries than a diet abundant in fat-soluble vitamins/low in phytic acid alone.
9	The Arrest of Dental Caries In Childhood	1928	J.D. Boyd, C.L. Drain [22]	Children under medical control and nutritional management for diabetes; N = 28	Human/ Retrospective Quasi-experimental	Eggs, butter, meat, cod liver oil, fruits, bulky vegetables, and 1 quart of milk and cream per day	After 6-months on the nutritionally managed diet, children exhibited arrest of their dental caries.
10	Dietary Control of Dental Caries ^b^	1929	J. D. Boyd, C.L. Drain C.M.V. Nelson [23]	Non-diabetic children from the orthopedic ward; N = 4	Human/ Quasi experimental	Followed the same nutritionally managed diet for diabetes as Boyd & Drain (1928): Eggs, butter, meat, cod liver oil, fruits, bulky vegetables, and 1 quart of milk and cream per day	Caries arrested after 2 months or longer; reverting from the nutritionally managed diet to the original diet for several months contributed to reemergence of caries in 2 children assessed.
11	Dietary Control of Dental Caries ^b^	1929	J. D. Boyd, C.L. Drain C.M.V. Nelson [23]	Preschool aged children with “actively progressive” caries N = 5	Human/ Quasi experimental	Diet included: “1 quart of milk, 1 egg, 1 teaspoon of cod liver oil, 1 ounce of butter, 1 orange, and one or more servings of succulent vegetables or fruits…”; candy after meals; followed at home	Caries arrested after following the diet for 10-weeks.
12	Dietary Control of Dental Caries ^b^	1929	J. D. Boyd, C.L. Drain C.M.V. Nelson [23]	Girls with Celiacs Disease N = 4 (2 older children had extensive caries; 2 younger children aged 25 and 31 months did not have caries)	Human/ Quasi experimental	Diet included: cod liver oil, orange or tomato juice, milk, vegetables and fruits daily; sugars and other simple carbohydrates; approximately twice as much protein as children in the prior groups and virtually no fat	The two younger children did not develop caries; the two older children exhibited arrested caries after following the dietary protocol for over 3 years.
13	Relation of Diet to General Health and Particularly to Inflammation of the Oral Tissues and Dental Caries	1930	M. T. Hanke [24]	Participants aged 0–50 years; N = 191; 4 groups: (1) no current dental disorders (*n* = 17); (2) caries (*n* = 61); (3) Gingivitis or pyorrhea and caries (*n* = 65); (4) no caries, but other conditions present (*n* = 48)	Human/ Quasi experimental	Vitamin C	All participants with caries were deficient in vitamin C; for *n* = 39 this was their only “demonstrable” nutritional deficiency
14	Remarks on The Influence of a Cereal-Free Diet Rich in Vitamin D and Calcium on Dental Caries in Children ^c^	1932	M. Mellanby, C.L. Pattison [25]	Puppies	Animal	Oatmeal; vitamin D	When oatmeal constitutes a significant portion of the diet it appears to contribute to “anti-calcifying effects” on teeth that are ameliorated with vitamin D.
15	Diet and Dental Health	1933	M. T. Hanke [26]	Institutionalized children aged 10–17 N = 323	Experimental	Orange juice; lemon juice	Among participants provided with a pint of orange juice and juice from 1 lemon per day for 1 year in addition to the standard institutional diet, 50% experienced arrest of their dental caries.
16	Retardation Of Dental Caries In Out-Patients Of A Dental Infirmary: Preliminary Study	1933	P.R. Howe, R.L. White, M. Rabine [27]	Children aged 2–11 who were patients at the Forsyth Dental Infirmary; N = 132; *n* = 104 cooperative with the diet; *n* = 28 not cooperative with the diet	Human/ Quasi-experimental	milk; raw and cooked vegetables, fruit (especially oranges), egg, meat, fish, butter on vegetables and bread; low cereals and breads; candy	Dental caries reduced by approximately 79% in just over one and a half years among those compliant with the diet.
17	The Production and Prevention of Dental Caries	1933	M.C. Agnew, R.G., Agnew, F.F. Tisdall [28]	Albino and hooded rats; 4-weeks old *n* = 365 fed a normal diet; *n* = 71 fed a diet low in phosphorus and vitamin D	Animal	Various, but focused on phosphorus, vitamin D	After 2–7 months, 70 of those fed a diet low in phosphorus and vitamin D developed caries; when adequate protein and other minerals were included, caries incidence was 50% at up to 13 months; no caries were evident in the group fed a normal diet.
18	The Influence of Vitamin D in the Prevention of Dental Caries	1934	P.G. Anderson, C.H.M. Williams, H. Halderson, C. Summerfeldt, R.G. Agnew [29]	Children aged 2–6 years in orphanages in Toronto Canada N = 162	Experimental	Vitamin D supplementation	Over 1-year, the group provided with vitamin D had less than half the number of new cavities than the group not supplemented with vitamin D; fewer “markedly progressive caries” were evident in the group supplemented with vitamin D; more “non-progressive caries” were evident in the vitamin D group.
19	The Improved Dentition of 5-Year-Old London School-Children	1944	M. Mellanby, H. Coumoulos [30]	Two separate cohorts of 5-year-old children in London, England in 1929 (*n* = 1293) and 1943 (*n* = 1604)	Retrospective multi-cohort	Public health changes that increased the availability of nutrient dense animal source foods (ASFs); fortification of bread with calcium and of margarine with vitamins A and D	Incidence of having “much” dental caries decreased from 62.8% to 29.3%, while the incidence of being “caries free” increased from 4.7% to 22.4% between 1929 to 1943.
20	The Reduction In Dental Caries In 5-Year-Old London School-Children (1929–47)	1948	M. Mellanby, H. Mellanby [31]	Two separate cohorts of 5-year-old children in London England in 1945 (*n* = 691) and 1947 (*n* = 1590)	Retrospective multi-cohort	Dietary public health changes spanning prenatal and postnatal periods of 5-year-old children.	A more rapid rate of increase in the percentage of “caries-free” or “almost caries-free” children was observed between 1945 and 1947 compared to 1943 and 1945 or 1929 and 1943.

Note: Only historic peer-reviewed studies are summarized in this table. ^a^ Original publication could not be obtained; results reported are those reported in Bunting (1931) [15]; ^b^ This report includes findings from several studies; ^c^ These findings are reviewed in Mellanby (1932) [25] but are from an earlier study (Mellanby, 1929) for which the title and article could not be obtained.

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
