# Peer review of "The Potential Systemic Role of Diet in Dental Caries Development and Arrest: A Narrative Review"

_nutrients, 2024, doi:10.3390/nu16101463_

Round 1

Reviewer 1 Report

Comments and Suggestions for Authors

It is a study that evaluates historical studies on diet and reinvestigates whether this historical knowledge can help inform current conceptualizations of tooth decay etiology.

The structure of the body presented by the authors is very hard to read. 

I suggest adding a table of the papers included in the analysis.

The headings did not follow any rationale. Some of them contain nutrient substances (e.g., Vitamins), and some of them are authors.

The authors should describe the concept of ASF consumption in the Introduction. This term appeared for the 1st time in the abstract and, after that, in the Discussion.

Additionally, in the introduction section, the authors should describe how to implement the conceptualizations of dental caries etiology based on diet to contemporary ecological theory. In the Discussion section, they should discuss and analyze that. 

Reviewer 2 Report

Comments and Suggestions for Authors

The manuscript presents a thorough review of the literature exploring the systemic impacts of diet on the development and arrest of dental caries, with a particular emphasis on the roles of fat-soluble vitamins and anti-nutrients. The paper is well-structured, offering a comprehensive overview of past and present research in this field and suggesting directions for future studies.

However, in terms of history, the authors need to revisit and integrate historical research into contemporary ones to facilitate the understandings of dental caries etiology. The manuscript does an exemplary job of summarizing a broad range of studies from the early 1900s, but lacks in the review of recent studies or perspective point of views.

1.    It would be beneficial for the authors to more critically assess their methodologies and potential biases. For example, a section discussing the limitations of historical studies and the need for modern research methodologies to confirm or refute these earlier findings could strengthen the paper.

2.    Providing more specific guidance on research questions, methodologies, or populations of interest could help to direct future studies in this area.

Round 2

Reviewer 1 Report

Comments and Suggestions for Authors

The authors have addressed the points. In the new PDF version, table 1 didn't appear. 

I have no further comment.

Author Response

Sorry about that. Please find attached the new PDF version of the manuscript with Table 1 attached. 
